# Does Multisystem Inflammatory Syndrome Only Mimic Acute Appendicitis in Children or Can It Coexist: When Should We Suspect MIS-C?

**DOI:** 10.3390/medicina58081101

**Published:** 2022-08-14

**Authors:** Idilė Vansevičienė, Ugnė Krunkaitytė, Inga Dekerytė, Mindaugas Beržanskis, Aušra Lukošiūtė-Urbonienė, Dalius Malcius, Vidmantas Barauskas

**Affiliations:** Pediatric Surgery Department, Lithuanian University of Health Sciences, Eivenių Str. 2, LT-50161 Kaunas, Lithuania

**Keywords:** appendicitis, MIS-C, multisystem, COVID-19, children, pediatric

## Abstract

*Background and Objectives*: Acute abdominal pain in children has been noticed to be a primary reason to seek medical attention in multisystem inflammatory disorder (MIS-C), which can prevail separately or together with acute appendicitis. Our aim was to distinguish regular appendicitis cases from MIS-C and to suggest the best clinical and laboratory criteria for it. *Materials and methods:* Cases of patients, admitted to the Pediatric Surgery Department over a six-month period in 2021, were retrospectively analyzed. Confirmed MIS-C or acute appendicitis cases were selected. MIS-C cases were either separate/with no found inflammation in the appendix or together with acute appendicitis. Acute appendicitis cases were either regular cases or with a positive COVID-19 test. Four groups were formed and compared: A-acute appendicitis, B-MIS-C with acute appendicitis, C-MIS-C only and D-acute appendicitis with COVID-19. *Results:* A total of 76 cases were overall analyzed: A-36, B-6, C-29 and D-5. The most significant differences were found in duration of disease A—1.4 days, B—4.5 days, C—4 days, D—4 days (*p* < 0.0001), C reactive protein (CRP) values A-19.3 mg/L B-112.5 m/L, C-143.8 mg/L and D-141 mg/L (*p* < 0.0001), presence of febrile fever A-13.9%, B-66.7%, C-96.6% and D-40% (*p* < 0.0001) and other system involvement: A 0%, B 100%, C 100% and D 20%. A combination of these factors was entered into a ROC curve and was found to have a possibility to predict MIS-C in our analyzed cases (with or without acute appendicitis) with an AUC = 0.983, *p* < 0.0001, sensitivity of 94.3% and specificity of 92.7% when at least three criteria were met. *Conclusions:* MIS-C could be suspected even when clinical data and performed tests suggest acute appendicitis especially when at least three out of four signs are present: CRP > 55.8 mg, symptoms last 3 days or longer, febrile fever is present, and any kind of other system involvement is noticed, especially with a known prior recent COVID-19 contact, infection or a positive COVID-19 antibody IgG test.

## 1. Introduction

The COVID-19 pandemic has been a difficult challenge to face globally with a lot of severe and fatal cases in adults. Children, however, seem to have been affected a lot less, making up less than 10 percent of all laboratory-confirmed cases, but with a possible untested higher prevalence, which was overlooked due to many children having milder symptoms, being asymptomatic or not being tested at all [1,2,3]. Children could be less affected due to not having comorbidities such as obesity, heart or chronic lung disease, diabetes or hypertension [4,5]. The ones who were more affected by the virus tended to be older children or teenagers, while data about younger children tends to be more conflicting [3,6,7]. In Spring of 2021, a new type of reaction to the virus was identified in children—multisystem inflammatory disorder (MIS-C), which presents 1–6 weeks after the initial infection and affects multiple different organs such as the skin, the heart, the lungs, the lymph nodes, the kidneys, the brain, and the gastrointestinal system [8,9]. While initially it was compared to Kawasaki disease, it has been shown to differ from it as well as the COVID-19 infection in adults [10,11,12]. Although MIS-C affects many different places in the body unlike COVID-19 itself, it appears to be less lethal; however, it should still be treated seriously because more than a half of the patients require intensive care and can develop lethal complications—such as the macrophage activation syndrome—as well as have long-term consequences, which have not yet been researched [13,14,15]. Due to a pronounced gastrointestinal presentation, MIS-C has been differentiated from acute appendicitis, as well as sometimes found to possibly prevail together. Even rarer are cases of confirmed COVID-19 with acute appendicitis cases, which currently have been described only in a few very rare instances in children.

Our aim was to compare and distinguish regular appendicitis cases from MIS-C and suggest the best clinical and laboratory criteria for it, as well as to see if COVID-19-positive appendicitis cases resemble MIS-C or are just regular acute appendicitis cases.

## 2. Materials and Methods

We retrospectively analyzed cases of patients, who were hospitalized with abdominal pain to the Pediatric Surgery Department at the Lithuanian University of Health Sciences Hospital over a six-month period in 2021 as well as selected cases with operated acute appendicitis or multisystem inflammatory disorder. MIS-C cases were either found to be separate (treated conservatively or surgically, but the appendix was found uninflamed) or with confirmed acute appendicitis (confirmed in histopathology, with the exceptions mentioned below when it was not possible). Regular acute appendicitis cases were found to be either separate or had a positive COVID-19 PCR Antigen test (without MIS-C criteria). The cases found were separated into the four following groups: Group A—patients with only acute appendicitis; group B—patients with acute appendicitis, who were also confirmed to have MIS-C; group C—patients with MIS-C without acute appendicitis; and group D—patients with acute appendicitis, who were found to be COVID-19 PCR Antigen positive (either initially or during the hospital stay). Surgically treated patients that had minimal or no inflammatory changes in the appendix and were diagnosed with MIS-C were categorized into group C.

We retrospectively analyzed their medical records and compared their demographic data (age, gender), duration of symptoms (in days), typical appendicitis symptoms, as well as MIS-C typical symptoms (other organ system involvement, febrile fever), laboratory tests on admission (C Reactive Protein (CRP), white blood cell (WBC) count, neutrophil percentage, platelet count and hemoglobin), ultrasound data as well as the pediatric appendicitis score (PAS) were counted for each patient. CT data was only interpreted in terms of respiratory system damage as a chest and abdominal CT was only performed for one patient in group B.

The stage of inflammation was taken from the histopathology report if the appendix was removed. Phlegmonous appendicitis was categorized as uncomplicated, while gangrenous or perforated was categorized as complicated. If surgical drainage due to an abscess or infiltrate was performed without removing the appendix (due to inability to remove it because of surrounding tissue damage), the stage of appendicitis was categorized as complicated. If an operation was performed but the appendix was found with minimal changes, which was confirmed on pathology report, it was not counted as acute appendicitis. A prior known positive test for COVID-19 PCR Antigen (in the last two months) was noted, as well as PCR COVID-19 antigen test results upon hospitalization or later until discharge and COVID-19 antibody IgG test results when the test was performed.

MIS-C was confirmed according to the WHO criteria: children/adolescents, with fever ≥3 days, at least two organ system involvement and evidence of COVID-19 or likely contact with COVID-19 infected patients [16], as decided by the physician treating the patient.

Statistical data analysis was performed by using IBM SPSS Statistics software. Quantitative data was found not to have a normal distribution using Kolmogorov and Smirnov tests, so a nonparametric Kruskal–Wallis test was used for the analysis. Results are described as median (minimum–maximum value). Qualitative data was compared using a Chi square for homogeneity criterion. Results are presented in absolute numbers and percentages. Obtained differences and relations were found significant if *p* < 0.05. Significant symptoms were added into a score, which was tested out using a receiver-operated characteristic curve (ROC), with a significant area under curve (AUC) > 0.5 when *p* < 0.05 was counted as significant and sensitivity/specificity was measured for the test.

## 3. Results

A total of 76 cases were overall analyzed for the study: 36 cases of operated acute appendicitis—group A; 6 operated acute appendicitis cases which later were confirmed to have MIS-C—group B; 29 cases with MIS-C (27 were treated conservatively, while 2 were treated surgically, but the inflammatory changes in the appendix were confirmed to be minimal or none on histopathology)—group C; and 5 patients surgically treated for acute appendicitis with a confirmed COVID-19 infection (either positive PCR antigen test on admission or later during hospitalization)—group D. We found that gender distribution was insignificant (presented below in Table 1), but solitary MIS-C patients tended to be the youngest with a median age of 8 years (1;15), while the oldest patients were in the COVID-19 positive group, with a median age of 13 years (2;15), *p* = 0.02. Symptoms lasted significantly less in group A than in all others, with a median of 1.4 (1;3) days vs. 4.5 days (group B) vs. 4 days (group C) vs. 4 days (group D), *p* < 0.0001.

The most common symptom overall was right lower quadrant pain, which had a high incidence in group A (86.1%), group B (100%) and group D (100%) but was mostly absent in group C (31%), *p* < 0.0001. Febrile fever was highly observed in both MIS-C groups—group B 66.7% and group C 96.6%—but was only present in 13.9% of group A cases, *p* < 0.0001. There were no significant differences in nausea, vomiting and diarrhea percentage, but both group A and D showed high rates of muscle rigidity (77.8% and 100%) and rebound tenderness (55% and 60%), while the MIS-C groups did not (muscle rigidity 50% in group B and 13.8% in group C, rebound tenderness 16.7% in group B and 6.9%in group C), *p* < 0.0001. Both MIS-C groups showed 100% of other system involvement (in both groups all patients had coagulation system disorders and commonly had cardiovascular changes 66.7% in group B and 75.9% in group C, with mucocutaneous symptoms being very prevalent only in group C 82.8%, as well as the respiratory system being affected in 48.3%), while acute appendicitis cases showed no other system involvement (but were not tested for specific disorders if they did not exhibit any symptoms), and group D had a proven coagulation disorder in one patient, who was tested; however, others did not have any symptoms and thus did not receive additional testing.

Abdominal ultrasound analysis showed that the appendix was significantly the least visualized in solitary MIS-C cases (group C) in only 28.6% (*p* < 0.0001) and had the lowest median diameter of the appendix 0.6 cm (0.4;1.3). Lymph node hyperplasia was present in a third of group B patients (33.3%) and group C patients (35.7%), while it was present in almost no group A or D patients (*p* = 0.011). Free fluid was not present in group B (0%) but was noticed in about half of group A patients (44.4%) and a little more than a half of group C patients (64.3%) and group D patients (60%), *p* = 0.03.

Laboratory tests showed that only group D cases had a positive PCR Antigen test, while 16.7% of cases in group B had a positive test prior to admission and 27.6% of cases in group C. All patients in both MIS-C groups who were tested for COVID-19 Antibody IgG had positive results. A very distinguishable and significant difference was noted in CRP levels: group A had the lowest median (19.3 mg/L (3;139.6)) in comparison to other groups (group B 112.5 mg/L (5;253), group C 143.8 mg/L (45;320) and group D 141 mg/L (36.2;196)), *p* < 0.0001. The lowest WBC (median 8.4 (3.2;23.7) × 10^9^/L) and platelet (168 × 10^12^/L (51;576)) counts were observed in MIS-C (group C) *p* = 0.004 and *p* < 0.0001, while the highest hemoglobin levels were found in the acute appendicitis group A—Hgb 136 (111;168) g/dL, *p* = 0.0001, but neutrophil percentage did not differ between the groups.

Upon analyzing these factors, we wanted to analyze if CRP alone and in combination with other factors could help to distinguish MIS-C (both with and without acute appendicitis) cases from acute appendicitis or acute appendicitis with COVID-19 in our data. We analyzed area under curve for C reactive protein to distinguish MIS-C cases from others (Figure 1). We found that CRP alone has an AUC = 0.9 with *p* < 0.0001, a sensitivity of 91.4% and specificity of 82.9% when it is more than 55.8 mg/L.

This has prompted us to see if a combination of CRP and other significant factors could improve these results. Upon closer inspection of previously analyzed data, we combined these factors into an MIS-C score (Table 2).

We applied this calculated score to our data in Figure 2. We found that it suggested even better predictability than CRP alone with an AUC = 0.983, with *p* < 0.0001 and a sensitivity of 94.3%, a specificity of 92.7% when at least three criteria were met.

## 4. Discussion

The importance of viruses in the pathogenesis of acute appendicitis has not yet been proven for certain. The possibility exists that viruses have influence either directly or indirectly, as cases where acute appendicitis was diagnosed with the presence of a virus have been presented in the literature. The viruses identified and mentioned are the Cytomegalovirus, Ebstein Barr virus, Rota virus, Adenovirus, Influenza virus, Varicella Zoster and even the Dengue virus [17,18,19,20,21,22]. However, these cases are sparse due to most patients with acute appendicitis usually not being tested for specific viral infections unless they have some specific viral symptoms, which clearly would not be present in regular acute appendicitis cases [17,19,23]. In healthy, non-immunocompromised children, the most probable cause of the virus affecting the appendix is hypertrophy of lymphoid tissue in the appendiceal wall, causing lumen obstruction, or viruses themselves residing in the lymphoid tissue and affecting epithelial cells of the appendix through various pathways [17,19,21,23,24,25]. However, it is also possible that some viral diseases and acute appendicitis share similar environmental or pathogenetic factors without any direct link [22].

The rise of COVID-19 with extensive testing has presented us with more possibilities than ever before to study the importance of a virus in the development of acute appendicitis. Although children make up the minority of laboratory confirmed cases with milder symptoms, this could be contributed to having a good early innate immune response to clear the virus [1,3,7,14]. It could also be explained by a possible lower maturity, functionality, and different distribution of the ACE2 receptor, which is the main SARS-CoV-2 viral entry site and is expressed in many organs of the body such as the lung, the heart, the ileum, the kidneys and the bladder [3,4,26]. Alas, even if COVID-19 itself rarely has a noteworthy presentation in children, the multiple inflammatory syndrome presenting 2–6 weeks after a SARS-CoV-2 infection requires prompt recognition and treatment [13,14,15,26,27]. Unlike the Kawasaki syndrome, which it has been compared to, MIS-C has been shown to differ in presentation having a more prominent gastrointestinal as well as cardiovascular involvement and affects slightly older children [9,12,14]. It also differs from COVID-19 itself as the respiratory system appears not to be as commonly affected [11,28,29].

The link to acute appendicitis has been put up for debate due to a very prominent gastrointestinal presentation in MIS-C [30]. Although some authors suggest that MIS-C can only mimic appendicitis and have shown successful conservative treatment, others claim that some cases can present together with acute appendicitis even in complicated, gangrenous or perforated form, where they require surgical treatment [10,27,31,32,33,34,35,36,37]. Some cases with complicated acute appendicitis were found to have viral SARS-CoV-2 RNA in both the appendix, the naso/oropharyngeal swab and serology [38]. There is a possibility that the appendix is affected just like the ileum, and in children, enterocytes could be a prime viral entry site through ACE2 receptors, rather than the respiratory tract [30,33,39]. It is also proposed that SARS-CoV-2 could act like a superantigen and may bind to host cells in a nonspecific manner, thus causing an excessive T cell activation (specifically Th1 subtype, which favors proinflammatory responses stimulating neutrophils, macrophages, causes vascular endothelial injury and matrisome activation) and inappropriate proinflammatory cytokine release [2,40,41]. T-cell responses have been found two times higher, although the total number of T cells and cytotoxic T cells has been found decreased, such as T reg cell responses in MIS-C [2,40,42]. MIS-C patients have also been found to have higher levels of IL-17α, IL-6, TNF- α, IL-1b, IFN-γ and IL-10 [14,43,44,45,46]. A similar cytokine pattern has earlier been described in cases of complicated appendicitis (IL-6, IL-17alfa and IL-10), as well as a more prominent Th1 inflammatory pattern [47,48,49]. All of these responses appear to be post infectious as most MIS-C patients have a negative SARS-CoV-2 PCR test [26], but all have positive antibodies IgG for COVID-19, which we also observed in our patients.

We have found both cases of MIS-C presenting with acute appendicitis, which required surgical treatment, as well as MIS-C cases where conservative treatment was sufficient. In our patients, we found that 17% (6/35) of all MIS-C cases in the year of 2021 were confirmed cases of acute appendicitis, of which 66.6% were complicated. On the other hand, we also had COVID-19 positive cases with acute appendicitis, with an 80% complicated appendicitis rate. High rates of complicated appendicitis could be linked to similar cytokine patterns discussed above.

We did not find any significant gender predominance; however, while acute appendicitis patients and MIS-C with appendicitis had a comparable median age, COVID-19 positive patients tended to be older, with a median of 13 years, and MIS-C patients without appendicitis were younger, with a median of 8 years. A very notable difference was found in terms of duration of symptoms: acute appendicitis cases had a median of 1.4 days duration and both MIS-C groups as well as the COVID-19 group had a much longer duration of symptoms of about 4 days. The long duration of symptoms could be explained by some patients seeking treatment earlier and having been prescribed antibiotics without referral to the hospital, but only ended up hospitalized and treated when the abdominal pain was more prominent, and the intensity and amount of symptoms increased with time. The need for intensive care in all MIS-C cases was about 45.7%, which confirms the numbers presented in the literature [13]. In terms of clinical symptoms, right lower quadrant pain has been found prominent in all appendicitis groups, except MIS-C without appendicitis, which only had a 31% rate, where other cases either lacked abdominal tenderness altogether or had a different localization of abdominal pain. COVID-19-positive appendicitis cases seemed to have the most prominent characteristics, similar to regular acute appendicitis cases, while only a half of patients in MIS-C with appendicitis had muscle rigidity but almost no rebound tenderness, and those cases without appendicitis were lacking in both. A prominent difference was found in terms of febrile fever, which was noted if it rose to 38 degrees Celsius at least once, where acute appendicitis cases only had a 13.9% history of febrile fever and MIS-C on the other hand with appendicitis had a 66.7%, without appendicitis 96.6% and COVID-19 positive appendicitis had 40%.

While regular acute appendicitis cases did not exhibit other system involvement, the same could be said about COVID-19 appendicitis cases, as MIS-C cases both with and without acute appendicitis show a high presence of cardiovascular involvement as mentioned in the literature [8]. We found not only elevated troponin levels but also cases with AV block, pericardial effusion, inotropic dysfunction, and coronary artery dilation, which confirms the data in other articles and prompts to always test the cardiovascular system in MIS-C [8,14,50,51]. Coagulation parameter changes were present in all MIS-C—even more prevalent than in COVID-19 cases mentioned in the literature, showing the importance of using antiaggregants such as aspirin to supplement treatment [11,52]. The most easily recognizable other system involvement without any testing has been found the mucocutaneous system, where a closer inspection of the patient might allow the physician to spot symptoms that should not be present in regular acute appendicitis cases (such as a skin rash, peripheral lymphadenopathy, strawberry tongue or conjunctivitis). Respiratory symptoms should be more common in COVID-19 cases, but we found that our patients (a third in MIS-C with appendicitis and half of patients in MIS-C) had incidence of decreased oxygen saturation, pulmonary infiltration or pleural effusion on X-ray or CT [29]. Neurologic symptoms have been few and in between with the most common being headaches or lethargy and only one patient experiencing seizures. In other author data similar neurocognitive symptoms are described as well as intracranial pressure elevation [36,53]. Urinary tract involvement was only seen in one patient with a bilateral pyelonephritis, however according to literature genitourinary involvement is more common in COVID-19 presentations rather than MIS-C [54].

Ultrasound has been described as unable to help differentiate between acute appendicitis and MIS-C due to a thickening of the appendix along with the distal ileum [33]. We found that groups differed in terms of free fluid, lymph node hyperplasia, as well as appendiceal visibility and diameter. MIS-C with appendicitis seems to present with the thickest appendiceal diameter of 1.3 cm and mesenteric lymph node hyperplasia in about a third of cases. MIS-C presents with mesenteric lymph node hyperplasia in a third of patients, but also the majority has free fluid visible and has a low appendix visibility of 28.6% with a lower diameter of 0.6 cm, which might explain the lower visibility. COVID-19 positive appendicitis cases seem to present more closely to the regular acute appendicitis cases with a comparable appendiceal diameter of 1 cm (vs. 0.9 cm in acute appendicitis cases) and most cases have free abdominal fluid and no mesentery lymph node hyperplasia.

Typical laboratory tests also had interesting outcomes. While Neutrophilia was present in different degrees and comparable in all groups, confirming other author data [46,55], we found that MIS-C with acute appendicitis had a similarly high white blood cell count on the initial blood test 13.8 × 10^9^/L (vs. 14.1 × 10^9^/L in acute appendicitis), but MIS-C cases without appendicitis had a normal white blood cell count 8.4 × 10^9^/L. C reactive protein levels were found high (>100 mg/L) in all patient groups except regular acute appendicitis cases, which could be explained by an intense inflammatory response, multiple organ system involvement in MIS-C, as well as simply a longer duration of symptoms—as CRP is not only inflammation degree, but very time dependent [56]. It is curiously different than CRP values in other cases of viral acute appendicitis cases, as they describe CRP values ranging from 5 to 50 mg/L [17,18,57]. Platelet count was also found significantly diminished only in the MIS-C without acute appendicitis with a median of 168 × 10^12^/L. These results are comparable to the literature, as thrombocytopenia and low WBC count have been used to differentiate from acute appendicitis, even with appendiceal lumen increase on ultrasound [35].

Upon seeing these differences, we selected a few noteworthy criteria to that could possibly help surgeons and pediatricians to understand how to suspect MIS-C in order to start treatment promptly. We found that if the patient suspected of acute appendicitis had three out of four criteria: CRP > 55.8 mg/L, duration of symptoms lasting three days or more, symptoms of other organ system involvement (even without prior testing such symptoms as a skin rash, conjunctivitis, peripheral lymphadenopathy and so on) as well as a history of a febrile fever of at least 38 degrees Celsius—he or she should be examined and tested for possible MIS-C, especially with a known recent prior COVID-19 infection, contact or a positive COVID-19 IgG test. It is important to mention that a patient with a perforated acute appendicitis might present similarly; therefore, we warrant to be cautious not to overlook this possibility and possibly still opt for surgical treatment when clear clinical, laboratory and ultrasound signs of acute appendicitis are present, as cases of solitary MIS-C appear to have less prominent abdominal symptoms as well as have a normal appendiceal diameter on ultrasound. Another possible problem with using our criteria is that gastrointestinal symptoms are usually the more prominent ones and other organ system involvement might be overlooked if the symptoms are mild and not tested for, and they sometimes tend to appear later during the progression of the disease. Fever also might not start on disease onset, but it develops in almost all cases if correct treatment is not applied, which is always insufficient in cases of MIS-C if they are only treated as acute appendicitis [58]. From clinical practice, we have noticed that suspicion of MIS-C should also arise when typical acute appendicitis cases do not seem to improve with treatment but rather deteriorate in terms of symptoms and inflammatory markers. COVID-19 positive cases seem to appear different from both regular acute appendicitis cases and those with MIS-C. While they share no other organ system involvement, which could just be mild and remain untested, they do appear clinically like complicated acute appendicitis cases with high CRP values. It is difficult to say if the severity of acute appendicitis is a direct result of the virus, as there are very few cases like this described in the literature [30,59].

## 5. Conclusions

We have studied and compared MIS-C cases both with and without confirmed acute appendicitis as well as COVID-19 positive acute appendicitis cases and compared them to regular acute appendicitis cases. We have found that MIS-C should be suspected even with clear acute appendicitis symptoms and laboratory markers when at least three out of four signs are present: CRP > 55.8 mg, symptoms last three days or longer, presence of febrile fever and any kind of other system involvement is noticed, especially with a known prior recent COVID-19 contact, infection or positive COVID-19 IgG antibody test. We advise to test for MIS-C but stress the importance of understanding that it can also present together with complicated forms of acute appendicitis. Cases of COVID-19 positive acute appendicitis appear to be both differ from MIS-C or separate acute appendicitis cases and are in need of more investigation.

## Figures and Tables

**Figure 1 medicina-58-01101-f001:**
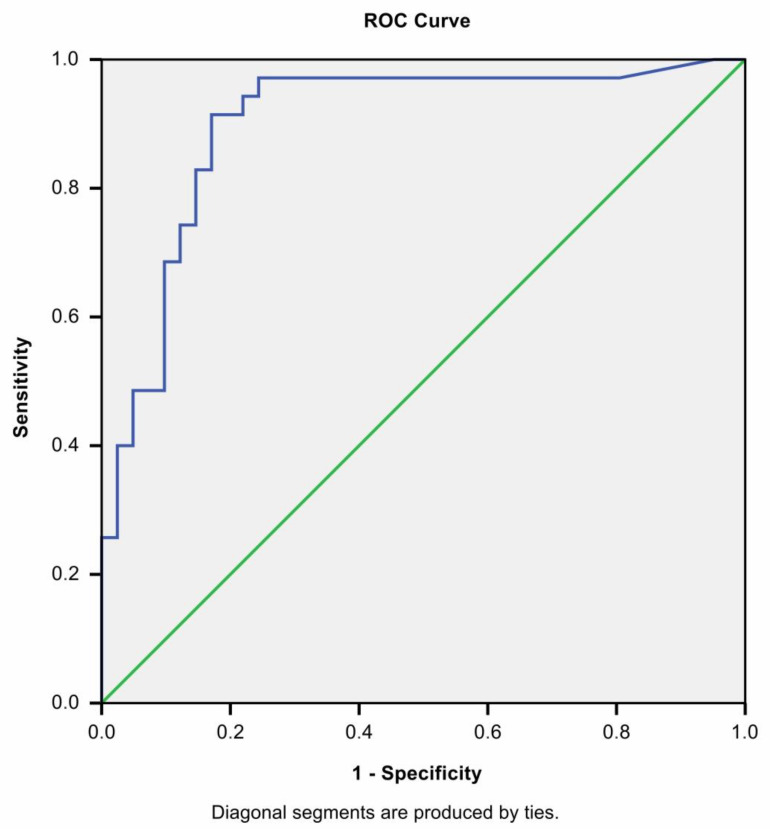
ROC curve of CRP when distinguishing MIS-C from AA cases.

**Figure 2 medicina-58-01101-f002:**
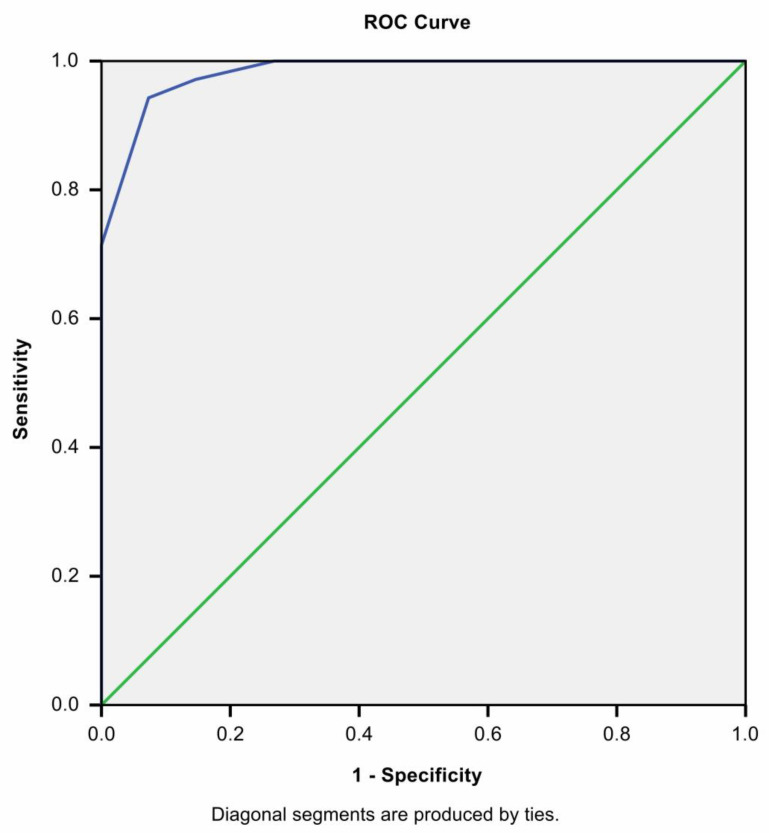
ROC curve of MIS-C score when distinguishing MIS-C from AA cases.

**Table 1 medicina-58-01101-t001:** Comparison of scute appendicitis (AA) cases and MIS-C cases.

	AA Group A	AA + MIS-C Group B	MIS-C Group C	AA + COVID-19 Group D	Significance (Group ABCD, Except Otherwise Stated)
Number of cases	36	6	29	5	
Gender M/F	50/50%	66.7/33%	65.5/34.5%	40/60%	*p* = 0.496
Age, years	11.5 (4;17)	10.5 (7;16)	8 (1;15)	13 (2;15)	*p* = 0.02
Symptom duration, days	1.4 (1;3)	4.5 (1;9)	4 (2;13)	4 (2;8)	*p* < 0.0001
Symptoms:					
RLQ pain	86.1%	100%	31%	100%	*p* < 0.0001
Nausea	75%	83.3%	58.6%	80%	*p* = 0.4
Vomiting	66.7%	66.7%	62.1%	60%	*p* = 0.977
Diarrhea	13.9%	50%	37.9%	40%	*p* = 0.078
Pain migration	44.4%	0%	0%	20%	*p* < 0.0001
Muscle Rigidity	77.8%	50%	13.8%	100%	*p* < 0.0001
Rebound tenderness	55%	16.7%	6.9%	60%	*p* < 0.0001
Febrile fever	13.9%	66.7%	96.6%	40%	*p* < 0.0001
Other organ system involvement:	0%	100%	100%	20%	*p* < 0.0001
* Mucocutaneous	0/36	2/6 (33.3%)	24/29 (82.8%)	0/5	*p* = 0.012 (Group B-C)
* Cardiovascular	0/36	4/6 (66.7%)	22/29 (75.9%)	0/5	*p* = 0.635 (Group B-C)
* Respiratory	0/36	2/6 (33.3%)	14/29 (48.3%)	0/5	*p* = 0.504 (Group B-C)
* Neurologic	0/36	2/6 (33.3%)	10/29 (34.5%)	0/5	*p* = 0.957 (Group B-C)
* Urinary	0/36	0/6	1/29 (3%)	0/5	*p* = 0.644 (Group B-C)
* Coagulation	0/36	6/6 (100%)	29/29 (100%)	1/5 (20%)	*p* = 1 (Group B-C)
Abdominal Ultrasound:					
Appendix visible	88.9%	83.3%	28.6%	100%	*p* < 0.0001
Appendiceal diameter, cm	0.9 (0.6;1.5)	1.3 (0.9;1.5)	0.6 (0.4;1.3)	1 (0.8;1.3)	*p* = 0.003
L/n hyperplasia	5.6%	33.3%	35.7%	0%	*p* = 0.011
Free fluid	44.4%	0%	64.3%	60%	*p* = 0.03
Laboratory tests:					
CRP, mg/L	19.3 (3;139.6)	112.5 (5;253)	143.8 (45;320)	141 (36.2;196)	*p* < 0.0001
WBC, ×10^9^/L	14.1 (5.4;30.8)	13.8 (8;21.1)	8.4 (3.2;23.7)	10.6 (10.2;11.2)	*p* = 0.004
Neutrophil %	82.9 (48.4;95.6)	75.1 (72.7;88.8)	84 (63;89.5)	73.3 (57.3;77.7)	*p* = 0.312
Platelet count, ×10^12^/L	270 (163;453)	320 (234;340)	168 (51;576)	253 (112;538)	*p* < 0.0001
Hemoglobin, g/L	136 (111;168)	127 (109;138)	120 (100;158)	120 (105;143)	*p* = 0.001
Histopathologic inflammation stage uncomplicated/ Complicated AA	50/50%	33/66.7%	-	20/80%	*p* = 0.145
PAS	6.5 (2;10)	7.5 (2;9)	4 (1;9)	7 (6;9)	*p* < 0.0001
Prior COVID-19 PCR Antigen positive test	0%	16.7%	27.6%	0%	*p* = 0.368
Covid PCR Antigen positive test (admission day or later)	0%	0%	0%	100%	*p* < 0.0001
COVID-19 IgG positive test	n/a	100% (when performed)	100% (when performed)	n/a	*p* = 1
Need for Intensive care	n/a	66.7%	41.4%	60%	*p* = 0.442

* Febrile Fever more or equal to 38 degrees Celsius; mucocutaneous symptoms: skin rash, peripheral lymphadenopathy, strawberry tongue, conjunctivitis; cardiovascular symptoms: inotropic dysfunction, arrhythmias, pericarditis or simply increased Troponin-I or Brain Natriuretic Peptide (BNP); respiratory symptoms: cough, low SpO2, pleuritis, infiltration on X-ray and so on; neurologic symptoms: seizures, lethargy, headaches; urinary symptoms: urea and creatinine increase, changes on urine test; coagulation disorders: changes in at least one coagulation parameter or increased d-dimers; PAS-pediatric appendicitis score. * Need for intensive care was found positive if at least one day was spent in the Pediatric Intensive Care Unit.

**Table 2 medicina-58-01101-t002:** MIS-C score when distinguishing from acute appendicitis.

Symptom/Test Result	YES	NO
CRP > 55.8 mg/L	1	0
Fever ≥ 38 °C	1	0
Other organ system involvement	1	0
Gastrointestinal symptom duration > 2 days	1	0

## Data Availability

The data presented in this study are available on request from the corresponding author.

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
