# Peer review of "Does Multisystem Inflammatory Syndrome Only Mimic Acute Appendicitis in Children or Can It Coexist: When Should We Suspect MIS-C?"

_medicina, 2022, doi:10.3390/medicina58081101_

Round 1

Reviewer 1 Report

although the topic may be interesting, I think that it has been extensively deepened in the last two years.

Moreover the paper needs a major revision of the English language.

Author Response

Thank you very much for your notes! We tried to our best to review and edit the English language mistakes such as typos, punctuation and so on. As recommended, we have also tried to find other relevant research data and supplement our article to represent more current data. We hope the improvement is noticeable.

Reviewer 2 Report

-In Abstract: You use the abbreviation MIS-C for the first time without the full term.

-In Introduction: revise the punctuation

-“Children could be 38 less affected due to not having comorbidities such as obesity, heart or chronic lung dis- 39 ease, diabetes or hypertension [4]” I recommend also this related reference to be  cited here (Rabaan AA, Al-Ahmed SH, Al Mutair A, et al. Immunopathogenesis and immunobiology of SARS-CoV-2. Le infezioni in medicina. 2021 Jun 1;29(2):167-80).

-In Methods: “MIS-C was confirmed according to the WHO criteria” Cite the reference and include it into the reference list.

- Histopathology examination (post operative) should be the accurate testing for inflamed appendix.

- Kindly mention the CT scan, if done for any patient or not performed.

- In results try to avoid redundancy.

-In Discussion: Sometimes you focus on repetition of the results instead of explanation or comparing your results, kindly discuss and explain results.

- Grammar and typo errors are detected (eg, muccodermal system,..), a careful language editing is advised

Author Response

Thank you very much for the much needed advice for improvement! First of all we have tried and corrected the grammar and typo mistakes. In the abstract we have used the full term of MIS-C as advised. We used the recommended reference in introduction and revised the punctuation. In methods the link was put into the reference list. Histopathology is the examination used in all patients, except where it was not possible to remove the appendix. But rather surgical drainage was performed due to extensive surrounding tissue damage due to infiltration. We have also mentioned the use of the CT scan (in methods) which was only used in 1 patient. We tried to improve the results as well as the discussion, adding newer data as well as trying to compare it to literature or explain it more where it was possible.